# Leptospirosis Incidence Post-Flooding Following Storm Daniel: The First Case Series in Greece

Irene Poulakida [1], Ourania S. Kotsiou [2], Stylianos Boutlas [1], Despoina Stergioula [1], Georgia Papadamou [1], Konstantinos I. Gourgoulianis [3] and Dimitrios Papagiannis [4,*]

1   Emergency Department, University Hospital of Larissa, 41500 Larissa, Greece; irene.poulakida@yahoo.com (I.P.); sboutlas@gmail.co (S.B.); deppyst@gmail.com (D.S.); georgia.papadamou@yahoo.com (G.P.)
2   Laboratory of Human Pathophysiology, Nursing Department, University of Thessaly, 41110 Larissa, Greece; raniakotsiou@gmail.com
3   Respiratory Medicine Department, School of Medicine, University of Thessaly, University Hospital of Larissa, 41500 Larissa, Greece; kgourg@uth.gr
4   Public Health & Adults Immunization Laboratory, Faculty of Nursing, School of Health Sciences, University of Thessaly, 41110 Larissa, Greece
*   Correspondence: dpapajon@gmail.com

**Abstract:** The present study investigates the public health impact of flooding on leptospirosis incidence after Storm Daniel in Thessaly, Greece, in September 2023. A notable increase in cases was observed, with seven cases of female patients and a mean age of 40.2 years, indicating a significant risk among working-age adults. From the end of September to the beginning of November 2023, a total of 35 patients from flood-prone areas presented to the Emergency Department of the Tertiary University Hospital of Larissa. Diagnosis of leptospirosis was established by meeting the criteria suggested by the national public health organisation (EODY)-compatible clinical course, epidemiological exposure, molecular and serologic confirmation by the detection of immunoglobulin M antibodies to leptospira spp. using a commercially available enzyme-linked immunosorbent assay and real-time quantitative PCR for the molecular detection of leptospira. The larger part (84.6%) of leptospirosis cases were associated with contact with floodwater. The majority of these patients (71.4%) were from the prefecture of Larissa, followed by 14.3% from the prefecture of Karditsa, 8.6% from the prefecture of Trikala, and 5.7% from the prefecture of Magnesia. Occupational exposure and urbanisation were key risk factors. The most prevalent clinical feature was rash (69.2%), followed by fever (61.5%) and myalgia (30.7%). The findings emphasise the need for robust public health strategies, improved sanitation, rodent control, and protective measures for sanitation workers. The data highlight the broader implications of climate change on public health and the necessity for ongoing surveillance and community education to mitigate future outbreaks.

**Keywords:** Storm Daniel; flooding; leptospirosis; occupational exposure; public health

## 1. Introduction

Leptospirosis is a significant global public health concern and is likely the most widespread zoonotic disease, particularly prevalent in tropical regions. It is a potentially fatal infectious disease caused by spirochete bacteria of the genus leptospira and represents a major direct zoonosis [1]. This contagious illness is caused by pathogenic leptospira, which can be transmitted directly or indirectly from animals to humans, though person-to-person transmission is rare.

Leptospires can be saprophytic, free-living bacteria that are generally non-pathogenic, or pathogenic with the potential to infect humans and animals [2]. Pathogenic leptospires are naturally maintained in various animal genital tracts and renal tubules. The disease

affects different organ systems to varying degrees, with symptoms ranging from asymptomatic or mild illness to severe disease that can lead to multi-organ failure and increased mortality [3].

Human infection occurs through contact with the urine of an animal reservoir or through contaminated water or soil. The bacterium thrives in humid environments, particularly where there is increased activity among ranchers or farm animals, as elevated humidity supports the pathogen's survival. Consequently, farmers and ranchers are at a higher risk of exposure due to their frequent contact with animals and the natural environment [3].

Floods impact millions of people annually [4], with extensive research indicating numerous health consequences, including increased mortality, injuries, poisonings, vector- and water-borne diseases, the exacerbation of chronic illnesses, and hunger. Indirect effects such as contaminated water, increased vector exposure, displacement, and overcrowding further elevate the risk of infectious diseases, skin infections, and vector-borne diseases [5–7].

Several studies have assessed the risk of leptospirosis in flood conditions, consistently finding a significant association between flood exposure and the occurrence of leptospirosis [8–12]. Factors such as age, gender, migration patterns, and occupation are critical risk determinants for leptospirosis. Historically, coal miners were the first occupational group identified as being at high risk for this disease [13]. The underreporting of leptospirosis is common due to a lack of clinical suspicion [14]. For the period 2004–2022 in Greece, the average annual incidence of leptospirosis cases was 0.22 cases per 100,000 population. The majority of patients (56.7%) had an agricultural occupation, and the region of the Ionian Islands showed the highest average annual frequency incidence of the disease (1.52 cases per 100,000). In 2022, three cases were recorded in Attica, and one case each in Thessaly, Epirus, and Eastern Macedonia and Thrace. According to the National Public Health Agency (EODY), the Thessaly region of Greece is classified as having an intermediate incidence of leptospirosis, with cases peaking during the summer months [15]. Official data indicate that cases of leptospirosis in Greece have been sporadic, with no documented series of cases reported to date (Figure 1). Previous studies in Greece from the Tertiary Referral Infectious Diseases Centre in Southwestern Greece have reported 45 confirmed cases of leptospirosis in Southwestern Greece, with an estimate incidence of approximately 1.1 cases per 100,000 population for a 4-year observation [16].

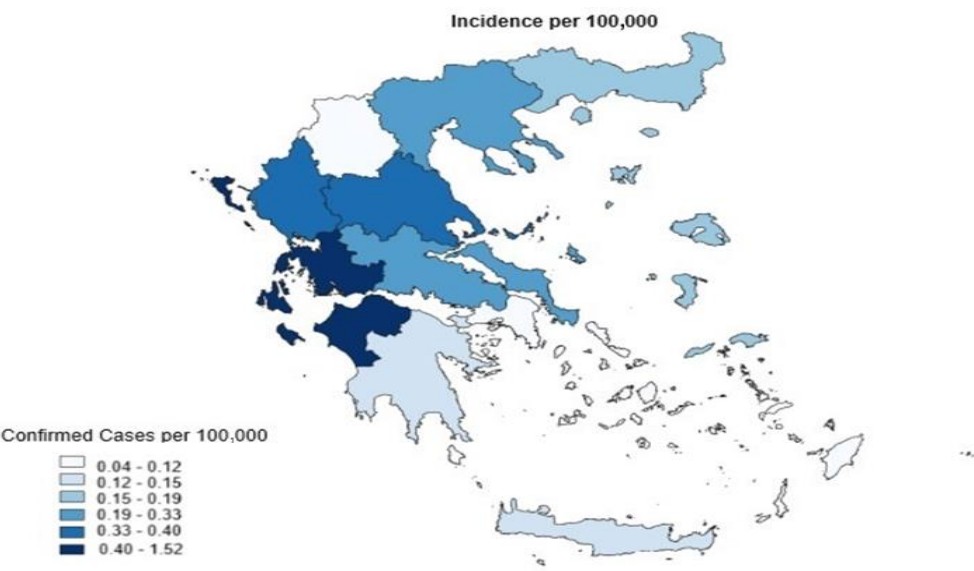

**Figure 1.** Average frequency of cases per region in Greece for the period 2004–2022. Source: EODY. Available at https://eody.gov.gr/wp-content/uploads/2023/09/leptospirosis_2004_2022.pdf (accessed on 27 June 2024).

The present study presents the first case series of leptospirosis in Greece for the two-month period which emerged following the devastating floods brought about by Storm Daniel.

## 2. Case Series Presentation

During the early autumn months of 2023, a significant increase in leptospirosis cases was observed in Thessaly, Greece, following a period of intense flooding. In response to the severe flooding in Thessaly in September 2023, the National Public Health Agency issued a strong advisory message, urging health authorities to maintain heightened vigilance for potential leptospirosis cases. According to the Greek public health organisation (EODY), the inclusion criteria were for symptomatic individuals who presented to the emergency department of the University Hospital of Larissa with a complaint of febrile illness (temperature > 38 °C), headache, myalgia, and physical weakness associated with any of the following symptoms—conjunctival suffusion/conjunctival haemorrhage, meningeal irritation, anuria or oliguria/proteinuria/haematuria, jaundice, haemorrhage, purpuric skin rash, or cardiac arrhythmia/failure—and who reported working in flood areas in Thessaly after the Storm Daniel floods. From the end of September to the beginning of November 2023, a total of 35 patients from flood-prone areas presented to the Emergency Department of the Tertiary University Hospital of Larissa with inclusion criteria and symptoms for leptospirosis. The majority of these patients (69.25%) were from the prefecture of Larissa, followed by 15.3% from the prefecture of Karditsa, 7.6% from the prefecture of Trikala, and 7.6% from the prefecture of Magnesia. A suspected leptospirosis case was clinically diagnosed based on WHO criteria, i.e., AUFI in patients (fever ≥ 38 °C) with headache and myalgia and a history of exposure to animal reservoirs or flooded environments. The laboratory confirmation according to EODY suggestions was the detection of pathogenic leptospira sp. by nucleic acid test (NAT) or a positive Leptospira (EIA) IgM result [17]. All patients underwent testing for leptospirosis through antibody detection and isolation using real-time qPCR. The criteria for serological testing for leptospirosis included patients from flood-prone areas who exhibited one or more of the following symptoms: fever, headache, chills, myalgia, conjunctival injection, rash, jaundice, or renal failure. Of the 35 patients tested for leptospirosis, 20 presented with fever and chills, 2 with kidney failure, 1 with jaundice, and 9 with rash. Additionally, most patients reported myalgia. Among these 35 patients, 13 were confirmed positive for leptospirosis based on compatible serological and molecular testing results. Only two of the examined patients required hospitalisation. The epidemiological data of these patients are presented in Table 1. We examined various factors, including their involvement in works and repairs following the floods, in both their homes and workplaces.

Exposure to a flooded environment was documented for all patients (100%). High-risk occupation or employment as the sole factor of exposure to the bacterium was identified in 2 of 13 confirmed cases: 15.3%. A significant number of patients engaged in high-risk activities, such as cleaning their homes or repairing public facilities. The majority (84.6%) of leptospirosis cases were associated with contact with floodwater, typically occurring while wading to inspect the surroundings of their homes or during the clean-up process. It is notable that the infections predominantly followed the flow of the floodwaters (see Figure 2). All cases were referred to EODY mandatory reference centre by our team. All actions and measures described in this report were part of an urgent public health response to the described outbreak. Personal data were handled according to existing processes and rules ensuring data protection and confidentiality, and all patients gave verbal consent about using the anonymised data for publication.

**Table 1.** Characteristics and factors of cases.

| Cases | Sex | Resident | Age | Symptoms | Laboratory Confirmation RT-PCR, IgM, IgG | Diagnosis | Exposure | Date | Region | Hospitalisation, Treatment with Antibiotics |
|---|---|---|---|---|---|---|---|---|---|---|
| 1 | Female | Rural | 43 | Myalgia, fever, rash | Yes | Leptospirosis | Work outdoors | 26 September 2023 | Larissa | Vibramycin |
| 2 | Female | Rural | 47 | Fever, jaundice, rash | Yes | Leptospirosis | Work outdoors | 24 September 2023 | Larissa | Vibramycin |
| 3 | Female | Urban | 26 | Fever, rash | Yes | Leptospirosis | Work outdoors | 26 September 2023 | Larissa | Vibramycin |
| 4 | Female | Rural | 22 | Skin rash, myalgia | Yes | Leptospirosis | Work outdoors | 28 September 2023 | Larissa | Vibramycin |
| 5 | Female | Rural | 76 | Fever, rash | Yes | Leptospirosis | Work outdoors | 21 September 2024 | Karditsa | Rocephin, 5 days of hospitalisation |
| 6 | Male | Urban | 61 | Rash | Yes | Leptospirosis | Work outdoors | 28 September 2023 | Larissa | Vibramycin |
| 7 | Female | Urban | 36 | Myalgia, kidney failure | Yes | Leptospirosis | Work outdoors | 10 October 2023 | Larissa | Rocephin, 11 days of hospitalisation |
| 8 | Male | Urban | 26 | Skin rash | Yes | Leptospirosis | Occupation | 10 October 2023 | Karditsa | Vibramycin |
| 9 | Male | Urban | 53 | Rash | Yes | Leptospirosis | Work outdoors | 10 October 2023 | Larissa | Vibramycin |
| 10 | Male | Urban | 32 | Fever, increasing CPK | Yes | Leptospirosis | Occupation | 12 October 2023 | Trikala | Vibramycin |
| 11 | Male | Rural | 49 | Fever, rash | Yes | Leptospirosis | Work outdoors | 16 October 2023 | Larissa | Vibramycin |
| 12 | Female | Rural | 27 | Hematoma, fever | Yes | Leptospirosis | Work outdoors | 14 October 2023 | Magnesia | Vibramycin |
| 13 | Male | Urban | 25 | Fever, myalgia | Yes | Leptospirosis | Work outdoors | 18 October 2023 | Larissa | Vibramycin |

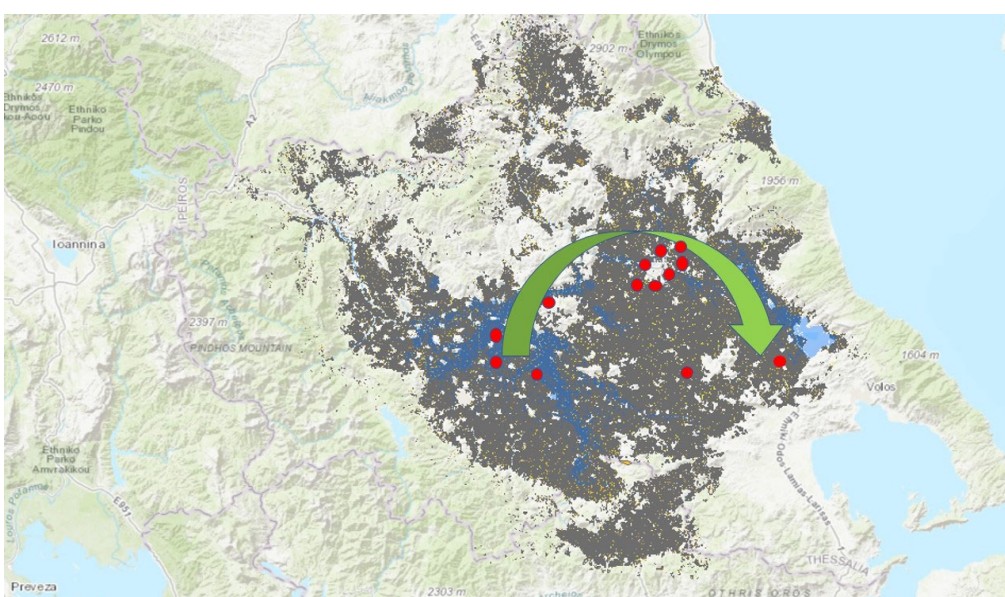

**Figure 2.** Geographical distribution of cases (red dots) of the infections according to flooded areas in Thessaly. The green arrow shows the direction of floods. Map of flooded area available at https://www.arcgis.com/home/webmap/print.html (accessed on 27 June 2024).

*Statistical and Laboratory Analysis*

Patient characteristics, including demographic details, laboratory findings, treatment modalities, and outcomes, were recorded in Microsoft Excel (Microsoft Corporation, Redmond, WA, USA). The continuous variables were presented as mean with standard deviation, and categorical variables were presented as numbers and percentages. A *p*-value of <0.05 was considered statistically significant. This study adhered strictly to the ethical guidelines outlined in the Declaration of Helsinki. All actions and measures described in this report were part of an urgent public health response to the described outbreak. Personal data were handled according to existing processes and rules ensuring data protection and confidentiality, and all patients gave verbal consent about the use of anonymised data for publication. The recent study was performed at the University Hospital of Larissa, a large academic institution, and one of the five hospitals in total in Thessaly which addressed the recent emergency public health issue after the Storm Daniel floods. A total of 35 patients were investigated. Serum and human blood were collected in sterile biological containers and processed for molecular detection/confirmation of Leptospira spp. The method used by the University Hospital of Larissa for serologic confirmation was an ELISA (Institut Viron Serion GmgH, Warburg, Germany) to check matched serum specimens for the presence of particular Leptospira immunoglobulin (Ig) M, as directed by the manufacturer. Three qualitative responses were obtained from the ELISA: positive, negative, and ambiguous (borderline positive/negative). Regarding PCR results, samples were characterised as positive or negative according to the detection of Leptospira DNA. Real-time qPCR was used for molecular diagnosis.

## 3. Discussion

Flooding is the most frequent natural disaster, and it significantly impacts both social and physical well-being. Post-flood environmental changes expose individuals to contaminated water supplies, sewage systems, and flood-prone areas, fostering the growth and spread of infections [18]. This study presents the first series of leptospirosis cases in Greece following the extensive flooding in Thessaly in autumn 2023. The engagement of the local population in cleaning up contaminated waters facilitated the spread of the disease to individuals not typically considered at high risk. However, the majority of these patients experienced mild symptoms that did not require hospitalisation.

According to data from the National Public Health Agency (EODY), leptospirosis cases in Greece have historically been more common among males, with males accounting for 82.2% of cases over the past 80 years [15]. In contrast, the current study found a slightly higher proportion of infections among females (53.8%). This divergence suggests potential shifts in exposure patterns and risk factors, warranting further investigation into gender-specific vulnerabilities and behaviours during post-flood activities.

The study observed a range of ages among the leptospirosis patients following the floods in Thessaly, Greece. The mean age of the patients was 40.2 years, with a standard deviation of 15.7 years. This indicates that leptospirosis affected a relatively young demographic: predominantly adults in their working years. In comparison to our report, a study supported by the World Health Organization reported that males are predominantly affected with an estimated 2–33 million DALYs or approximately 80% of the total burden of leptospirosis [19]. Age is a significant factor in the epidemiology of leptospirosis. Younger adults may be more likely to engage in activities that increase their exposure to contaminated water, such as participating in cleanup efforts or working in occupations that involve direct contact with floodwaters. Age also is an important factor for the severity of leptospirosis. In endemic areas, leptospirosis is more common and more severe in adults compared with children. A study was conducted in New Caledonia for a six-year period and found that the frequency of several classic severe disease manifestations was significantly lower among small children compared with adolescents [20]. Furthermore, the age distribution in this study aligns with global patterns in which leptospirosis is often seen in economically active individuals who are more likely to be engaged in outdoor activities or occupations that involve contact with water or animals. The relatively young age of the patients highlights the importance of targeted public health interventions for this demographic to reduce the risk of infection. The findings emphasise the need for age-specific health education and preventive measures. Younger adults should be particularly cautious and equipped with protective measures during and after flood events to mitigate their risk of contracting leptospirosis. Public health campaigns should focus on informing this age group about the dangers of exposure to contaminated water and the importance of using appropriate protective gear.

Exposure to polluted environments after floods can lead to disease transmission regardless of protective measures. Human infections can result from exposures at work, during leisure activities, or through amateur endeavours. Occupation is a major risk factor, with most leptospira infections found among farmers, veterinarians, and abattoir workers, resulting from direct contact with infected animals. Indirect contact is particularly relevant for sewage workers, miners, soldiers, septic tank cleaners, fish farmers, gamekeepers, canal workers, and rice field workers [21–23]. In this study, two cases (15.3%) involved high-risk occupations, one worker from the public electricity company and another from a construction company, both of whom reported symptoms following exposure to floods while repairing facilities.

Urbanisation has been linked to an increased risk of leptospirosis, likely due to environmental factors that enhance the persistence of the bacteria and facilitate shared pathways of transmission. Studies have shown that urban and rural areas both face risks, primarily due to variations in rat populations and exposure to mud flows and flooding [24]. The current study indicates a balance between rural and urban cases, with an increase in urban cases compared to previous data that predominantly reported rural cases. This shift suggests that leptospirosis is emerging in metropolitan areas, reflecting trends observed in other industrialised countries, particularly in Europe [25].

The spread of leptospirosis in Thessaly following the floods highlights the importance of environmental and behavioural factors in disease transmission. It is notable that the first recorded case of the present study, the No. 5 case from Karditsa, and the case from Magnesia, No. 12, showed that the infections followed the flood direction to drainage basins. Contaminated water and inadequate sanitation infrastructure in flood-affected areas can exacerbate the risk of leptospirosis. This study emphasises the need for compre-

hensive public health strategies that address these environmental challenges, alongside promoting safe practices during and after flood events. Effective public health interventions must include community education, improved sanitation, rodent control, and protective measures for individuals involved in cleanup activities.

The findings of this study underscore the necessity for ongoing surveillance, research, and preparedness to manage leptospirosis and other infectious diseases in flood-prone regions. Early detection and prompt responses are critical in mitigating the health impacts of flooding. The rise in leptospirosis cases following the Thessaly floods highlights the need for heightened awareness and preventive measures among health professionals and the community. Future research should explore the long-term effects of flooding on public health and evaluate the effectiveness of various intervention strategies to build resilience against similar threats. The present study suffers from several limitations, firstly due to the design of the study and the small number of participants, and secondly due to the nature of our department; important clinical haematological or clinical biochemistry data were missing which could further support our manuscript.

## 4. Conclusions

This study highlights the significant public health impact of flooding on leptospirosis incidence, particularly following the severe floods caused by Storm Daniel in Thessaly, Greece, in September 2023. The findings demonstrate that extreme weather events can drastically alter disease dynamics, increasing risks, especially during post-disaster clean-up. Notably, there was a shift in gender distribution, with a higher proportion of female patients compared to historical data, suggesting changes in exposure patterns or susceptibility. Occupational exposure also played a critical role, with high-risk jobs contributing to the disease's incidence. This study indicates that leptospirosis is emerging as a concern in urban as well as rural areas, necessitating comprehensive urban health strategies. Prompt public health responses and continuous surveillance are essential in managing outbreaks. The Thessaly data underscore the broader implications of climate change on public health, highlighting the need for robust public health infrastructure and community education to build resilience against future leptospirosis outbreaks.

**Author Contributions:** I.P., S.B., D.S. and G.P. contributed to data collection. Statistical analyses were performed by I.P. and D.P. Data analysis: I.P., O.S.K. and D.P. Drafting the manuscript: I.P., S.B., O.S.K. and D.P. Supervising the study: D.P., G.P. and K.I.G. All authors participated in the revision of the manuscript. All authors have read and agreed to the published version of the manuscript.

**Funding:** This research received no external funding.

**Institutional Review Board Statement:** Ethical review and approval were waived for this study due to all actions and measures described in this report were part of an urgent public health response to the described outbreak. Personal data were handled according to existing processes and rules ensuring data protection and confidentiality. This report does not contain elements linking data to specific events or persons and thus an ethics committee approval was not sought.

**Informed Consent Statement:** Informed consent was obtained from all subjects involved in the study.

**Data Availability Statement:** Data will be made available upon reasonable request by accredited researchers.

**Conflicts of Interest:** The authors have no competing interests.

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
