# Peer review of "Leptospirosis Incidence Post-Flooding Following Storm Daniel: The First Case Series in Greece"

_2036-7449, doi:10.3390/idr16050069_

Round 1

Reviewer 1 Report

Comments and Suggestions for Authors

In this manuscript, Poulakida et al. report an outbreak of leptospirosis in Greece after flood events. As we have little information on the epidemiological situation of leptospirosis in this country, this study is interesting. However, the authors should indicate their case definition.

Major comments:

Authors should describe how is done the surveillance of leptospirosis in Greece (reference center ? mandatory ?)

Authors mention 35 patients, including 13 confirmed by serology and/or PCR. Please describe the methods used (ELISA IgM, MAT, PCR) and the case definition used in this study (probable / confirmed cases). what about the 22 other cases not confirmed by laboratory diagnosis, why are they included in the leptospirosis cases ?

Authors claim a higher proportion of females but among the laboratory confirmed cases this corresponds to only 7 females/ 13 cases; sampling is small and non significant.

resolution of figures is low

they are some redundancy in the introduction: for example lanes 32-33 and 41-42.

Table 1: please mention the sampling date and prefecture for each case to show they all belong to the same (or different) clusters. For laboratory confirmation, please indicate which methods was used.

Comments on the Quality of English Language

good level of english

Author Response

Please find attached the point by point response to reviewers comments. We would like to thank you for your constructive comments.

Reviewer 2 Report

Comments and Suggestions for Authors

Is there any data showing the floodwater had increased vector levels. This data would strengthen the basis of the article.

Lines 102-103: Can you clarify what does 15.3% indicate? Is it 15.3% of cased had high-risk occupation as sole factor of exposure?

I see "two of cases": can you clarify 2 our of how many cases?

Author Response

(The authors gave the same response as above.)

Reviewer 3 Report

Comments and Suggestions for Authors

Kindly refer to the attached report.

Thank you,

Comments on the Quality of English Language

Kindly refer to the attached report

Author Response

(The authors gave the same response as above.)

Reviewer 4 Report

Comments and Suggestions for Authors

The authors present a complete description of leptospirosis cases in Greece related to storm Daniel in September 2023. There are some defects of form in the table, which are in any case minor. Otherwise, I have no major corrections to make.

Author Response

(The authors gave the same response as above.)
